- 1 Dependence of the hygroscopicity parameter  $\kappa$  on particle size,
- 2 humidity and solute concentration: implications for laboratory
- 3 experiments, field measurements and model studies
- 4 Zhibin Wang<sup>1</sup>, Yafang Cheng<sup>1,2,\*</sup>, Nan Ma<sup>1,3</sup>, Eugene Mikhailov<sup>4</sup>, Ulrich Pöschl<sup>1</sup>,
  5 Hang Su<sup>2,1,\*</sup>
- 6 <sup>1</sup>Multiphase Chemistry Department, Max Planck Institute for Chemistry, Mainz
- 7 55020, Germany
- 8 <sup>2</sup>Institute for Environmental and Climate Research (ECI), Jinan University,
- 9 Guangzhou 511443, China
- 10 <sup>3</sup>Leibniz-Institute for Tropospheric Research, Leipzig 04318, Germany
- <sup>4</sup>St. Petersburg State University, 7/9 Universitetskaya nab, St. Petersburg 199034,
- 12 Russia
- 13 \*Correspondence to: Yafang Cheng (yafang.cheng@mpic.de) and Hang Su
- 14 (h.su@mpic.de)
- 15 Journal: Atmospheric Chemistry and Physics

# 16 Abstract

17 The hygroscopicity parameter  $\kappa$  has been intensively used in the investigation of 18 the water uptake, cloud condensation nuclei (CCN) activity and chemical 19 composition of atmospheric aerosol particles. A representative value of  $\kappa$  is often 20 assigned to individual species or sources. Such treatment may lead to confusion in 21 closure studies of  $\kappa$  derived from hygroscopic growth factor measurements ( $\kappa_{gf}$ ) and 22 CCN activity measurements ( $\kappa_{CCN}$ ), and in studies of aerosols at the sub-10 nm size 23 range. Here we show that for particles of the same dry composition,  $\kappa$  may differ as a function of water content, solute concentration and particle size. The concentration-24 25 and size-dependence of  $\kappa$  are demonstrated for representative inorganic and organic 26 compounds, i.e., ammonium sulfate (AS), sodium chloride (NaCl) and sucrose. Our 27 results illustrate that an absolute closure between  $\kappa_{gf}$  and  $\kappa_{CCN}$  should not be expected, 28 and how the deviations observed in field and laboratory experiments can be 29 quantitatively explained and reconciled. The difference between  $\kappa_{\rm gf}$  and  $\kappa_{\rm CCN}$ 30 increases as particle size decreases reaching up to 40% and 30% for 10 nm AS and 31 NaCl particles, respectively. Moreover, we show that the deviations of  $\kappa_{CCN}$  vary from 32 ~10% for 30 nm and ~40% for 200 nm, indicating a strong dependence on the Köhler 33 models and thermodynamic parameterizations used for instrument calibration (e.g., 34 effective water vapor supersaturation in CCN counter). By taking these factors into 35 account, we can largely explain apparent discrepancies between  $\kappa_{\rm gf}$  and  $\kappa_{\rm CCN}$  values reported in the scientific literature. Our results help to understand and interpret  $\kappa$ 36 37 values determined at different water vapor ratios and at different size ranges (especially sub-10 nm). We highlight the importance of self-consistent 38 39 thermodynamic parameterizations when using AS for calibration aerosol and taking it 40 as a reference substance representing inorganics in closure study between chemical 41 composition and hygroscopicity of aerosol particles.

## 42 1. Introduction

Hygroscopicity and cloud condensation nuclei (CCN) activity represent the 44 ability of aerosol particles to interact with water, which is essential for the 45 understanding of aerosol climate effects (Andreae and Rosenfeld, 2008). These 46 properties can be described by the Köhler theory (Köhler, 1936), which accounts for 47 both Kelvin and solute effects. Water activity and surface tension are two key 48 parameters in the Köhler equation, both of which are functions of the aerosol 49 composition and solution concentration. The concentration dependence of water activity and surface tension has been determined for many compounds, e.g., 50 51 ammonium sulfate (AS) and sodium chloride (NaCl) etc. (Tang and Munkelwitz, 52 1994; Tang, 1996; Pruppacher and Klett, 1997). Yet we are in lack of such information for a large number of aerosol species and mixtures. 53

To describe the relationship between particle dry diameter and CCN activity, 55 Petters and Kreidenweis (2007) proposed a method using a single hygroscopicity 56 parameter  $\kappa$ . The  $\kappa$  parameter has several advantages: (1) representative values of  $\kappa$ 57 may be assigned to a specific aerosol species or source; (2) values of  $\kappa$  for mixtures 58 may be determined from volume-weighted average  $\kappa$  of individual components (Petters and Kreidenweis, 2007); and (3) the experimentally-determined  $\kappa$  has already 59 60 accounted for the impacts of aerosol size, composition and surfactants (Facchini et al., 61 1999). This approach has been proved useful in describing and predicting the CCN activity of single components and aerosol mixtures (Farmer et al., 2015, and 62 references therein), and the simplified parameterization of  $\kappa$  have been implemented 63 in cloud modeling studies (Spracklen et al., 2008; Reutter et al., 2009; Pringle et al., 64 65 2010; Chang et al., 2015).

The original purpose of introducing  $\kappa$  is to achieve a simple prediction of critical 67 activation dry diameter and supersaturation for the CCN activation of aerosol

particles (Petters and Kreidenweis, 2007). Since representative  $\kappa$  value is assigned to 69 an aerosol species, it is legitimate to ask if the experimentally determined  $\kappa$  can be 70 used to estimate the aerosol composition. Such efforts have been encouraged by the 71 finding of a simple near linear relationship between hygroscopicity parameter  $\kappa$  and 72 the organic mass fraction (Petters and Kreidenweis, 2007; Gunthe et al., 2009), which 73 has been confirmed by results from different locations and sources (Dusek et al., 2010; 74 Cerully et al., 2011; Gunthe et al., 2011; Kawana et al., 2016; Vogel et al., 2016). In 75 addition, the change of  $\kappa$  values was regarded as an indicator for the evolution of 76 chemical composition or mixing state in the aging process of aerosol particles (Chang 77 et al., 2010; Massoli et al., 2010; Wang et al., 2010; Alfarra et al., 2013; Lathem et al., 78 2013; Mei et al., 2013a; Mei et al., 2013b; Zhao et al., 2015), and the size-dependent 79  $\kappa$  values have also been used as an evidence of size-dependent chemical compositions 80 (Su et al., 2010; Cerully et al., 2011; Rose et al., 2011; Lance et al., 2013).

The Köhler equation describes not only the CCN activation under supersaturated 82 conditions but also the hygroscopic growth under subsaturated conditions. 83 Accordingly,  $\kappa$  values can also be determined from hygroscopic growth data 84 measured by hygroscopic tandem differential mobility analyzer (HTDMA). Hence 85 closure studies have been performed to compare  $\kappa$  values determined by hygroscopic 86 growth factor measurements ( $\kappa_{\rm sf}$ ) to those determined by CCN activity measurements 87 ( $\kappa_{CCN}$ ). Instead of a closure, most laboratory (Carrico et al., 2008; Massoli et al., 2010; 88 Dusek et al., 2011; Alfarra et al., 2013; Hansen et al., 2015; Dawson et al., 2016) and field measurements (Good et al., 2010; Irwin et al., 2010; Cerully et al., 2011; Wu et 89 90 al., 2013; Bougiatioti et al., 2016; Kim et al., 2016) showed that  $\kappa_{CCN}$  is usually larger 91 than  $\kappa_{gf}$ . Wex et al., (2009) demonstrated that the constant  $\kappa$  value over different 92 concentrations should be reconsidered due to the non-ideality effects in the solution 93 droplet and surface tension variation. Especially in mixed particles, the organic 94 coating or the presence slightly soluble substances can reduce the water transport

across the surface by acting as a physical barrier, which may lead to a discrepancy by 96 a factor of 5–10 between  $\kappa_{\rm CCN}$  and  $\kappa_{\rm gf}$  for secondary organic aerosol (Petters et al., 97 2009; Hansen et al., 2015; Mikhailov et al., 2015; Pajunoja et al., 2015). Hersey et al., 98 (2013) found that the aerosol aging and biomass burning can lead to the discrepancies 99 between  $\kappa_{\rm CCN}$  and  $\kappa_{\rm gf}$  values that may be related to mixing state. Conversely, higher 100  $\kappa_{\text{gf}}$  compared to  $\kappa_{\text{CCN}}$  was also observed, and the discrepancies were more attributed 101 to the instrument technical differences (Irwin et al., 2011). Moreover, recent studies 102 (Keskinen et al., 2013; Kim et al., 2016) tried to related the 103 experimentally-determined  $\kappa$  value to chemical composition during new particle 104 formation and initial growth stage, based on the  $\kappa$ -composition relationship built for 105 much larger particles (dry diameter larger than ~50 nm). Wang et al. (2015), however, 106 reported much smaller  $\kappa$  values for the sub-10 nm aerosols than to the hundred 107 nanometer particles with the same chemical composition. A question is hence raised, 108 if and how accurate a single  $\kappa$  value can be used to represent the whole K öhler curve 109 for the whole dry size range. The answer to this question is especially critical and 110 sensitive for the understanding of multiphase chemistry (Herrmann et al., 2015; 111 Pöschl and Shiraiwa, 2015; Cheng et al., 2016) and for accurate climate modeling 112 (Pajunoja et al., 2015).

Another critical issue concerning the application of  $\kappa$  is that its exact value is 114 subject to the selection of thermodynamic parameterizations in K öhler models (Rose 115 et al., 2008; Mikhailov et al., 2013). This is because different thermodynamic 116 parameterizations may lead to different K öhler curves (Cheng et al., 2015; see Fig. 1 117 and references therein), resulting in different calibrations of effective water vapor 118 supersaturation ( $S_e$ ) in CCN counter (CCNC) and column relative humidity (RH) in 119 HTDMA and consequently different  $\kappa$  values.

In this study, we address the question whether the same compound always has 121 the same  $\kappa$  value; if not, what contributes to the difference. In the following, we first

present the theoretical basis and data retrieval methodology. Then we demonstrate the concentration and size dependence of  $\kappa$  for exemplary substances. Finally, we discuss the closure studies between  $\kappa_{gf}$  and  $\kappa_{CCN}$ , and emphasize the importance of the usage of a consistent K öhler model for both calibration with AS and the retrieval of AS  $\kappa$ value when taking it as a reference substance in the inorganic/organic mixing ratio analyses.

# 128 2. Methodology

#### 129 2.1 к-К öhler model

The K öhler theory describes the equilibrium saturation ratio (*s*) over a spherical
aqueous droplet (K öhler, 1936):

$$s = a_{\rm w} \exp(\frac{4\sigma_{\rm sol}v_{\rm w}}{RTD_{\rm d}gf}) \tag{1}$$

where  $a_w$  is the water activity,  $\sigma_{sol}$  is the surface tension of solution droplet,  $v_w$  is the partial molar volume of water,  $D_d$  is the particle dry diameter (mass equivalent diameter), *gf* is the growth factor of particle diameter relative to the dry particle diameter (*gf* =  $D/D_d$ ), which is related with solute concentration by concentration-dependent solution density. *R* and *T* are the universal gas constant and absolute temperature, respectively.

For AS and NaCl aerosols, we choose the original models used by Biskos et al. (2006a; 2006b), which agree well with the observation data from HTDMA experiments (Cheng et al., 2015).  $a_w$  and  $\sigma_{sol}$  are expressed as a function of solute mass fraction  $x_{s}$ , which are derived from the best fit to the literature measurement data (Tang and Munkelwitz, 1994; Tang, 1996; Pruppacher and Klett, 1997). The parameterizations are detailed in the Appendix.

Petters and Kreidenweis (2007) proposed a hygroscopicity parameter  $\kappa$  for the 146 parameterization of water activity:

$$\frac{1}{a_{\rm w}} = 1 + \kappa \frac{V_{\rm s}}{V_{\rm w}}$$
(2)

where  $V_{\rm s}$  and  $V_{\rm w}$  are the volumes of the dry aerosol particle and water, respectively. 149 For ideal solution,  $\kappa$  have a unique value for certain species while it may vary for 150 non-ideal solution.

Substituting the expression of  $\kappa$  into Eq. (1), we have the  $\kappa$ -K öhler equation as:

$$s = \frac{gf^3 - 1}{gf^3 - (1 - \kappa)} \exp(\frac{4\sigma_{\rm sol}v_{\rm w}}{RTD_{\rm d}gf})$$
(3)

Due to the lack of thermodynamic data for the mixed or unknown system, 154 surface tension of water ( $\sigma_w$ ) is used instead of  $\sigma_{sol}$  and  $v_w$  is simplified as  $M_w/\rho_w$  in 155 the  $\kappa$  calculation (Petters and Kreidenweis, 2007; Su et al., 2010), where  $M_w$  and  $\rho_w$ 156 are the molar mass and density of water, respectively.

# 157 **2.2 Determination of** *κ*

To elucidate the concentration and size dependence of  $\kappa$ , we use AS, NaCl and sucrose as exemplary substances. The 'real' *s-gf* relations for these substances at different sizes (i.e.,  $D_d$ ) are first calculated by Eq. (1) with well-documented thermodynamic data (i.e.,  $a_w$  and  $\sigma_{sol}$ ). We then retrieve the corresponding  $\kappa$  values for each s-*gf*- $D_d$  pair with Eq. (4):

$$\kappa = \frac{gf^3 - 1}{s} \exp(\frac{A}{D_d gf}) - gf^3 + 1 \text{ and } A = \frac{4\sigma_w M_w}{\rho_w RT}$$
(4)

# 164 **3. Results and discussion**

#### 165 **3.1 Concentration and size dependence of** $\kappa$

In reality, the aerosol or cloud droplets are often not ideal solutions, especially at concentrated state, and the Zdanovskii-Stokes-Robinson (ZSR) volume additivity 168 169 assumption would not hold anymore. To maintain the  $\kappa$ -K öhler equation (Eq.3), these 170 non-ideality effects have to be compensated, which is reflected by a change in  $\kappa$ . That 171 is why a single species does not correspond to a unique  $\kappa$  value. In addition, since  $\sigma_{\rm w}$ 172 is often used instead of  $\sigma_{sol}$ , their difference also needs to be compensated through a 173 change in  $\kappa$  values. In a word, the  $\kappa$  in practical application is subject to both the 174 non-ideality effect and simple treatment of surface tensions. The nano-size effects 175 (e.g., the Tolman effect) on the thermodynamic properties of aerosol particles may 176 result in additional size dependence as well (Cheng et al., 2015).

**Concentration dependence.** Figure 1 shows the  $\kappa$  values as a function of solute 178 concentration (expressed as gf or solute molality  $\mu_s$ ) for different aerosol particles with dry diameters of 10 nm, 50 nm and 100 nm, respectively. Here the change of gf 179 180 reflects the change of solution concentration, i.e., larger gf corresponds to more dilute 181 solution (see Eq. A5). Our theoretical calculations illustrate a strong concentration 182 dependence of  $\kappa$  values for AS, NaCl and sucrose. Similar findings in sub-saturation 183 range (RH

- 2007). However,  $\kappa$  of sucrose aerosols has a different concentration dependence
- which shows a monotonic decrease down to ~0.08. Compared to the 100 nm aerosols,
- the 10 nm and 50 nm aerosols show similar concentration dependence but lower  $\kappa$ 197 values.
- Size dependence. Beside the concentration dependence, Fig. 1 also shows a size 199 dependence of  $\kappa$ . This feature can be more clearly seen in Fig. 2, where  $\kappa$  values are 200 plotted against the particle diameter. The same color-coding represents the same 201 solution concentration, which helps to separate the size effect from the concentration 202 effect. Given the same gf, the  $\kappa$  values increase with increasing particle diameter and 203 finally approaching a plateau, i.e., ~0.56 for AS (gf~1.5), ~1.8 for NaCl (gf~2.0) and 204 ~0.14 for sucrose (gf ~1.2), respectively. The size dependence of  $\kappa$  is more prominent 205 for AS aerosols, especially for smaller particles. For the same concentration (gf of  $\sim$ 206 1.5), the  $\kappa$  values of AS vary from 0.40 for 6 nm to 0.51 for 20 nm, and then slowly 207 grow to 0.56 for 300 nm.

# 208 3.2 Comparison of HTDMA and CCNC measurements

#### 209 3.2.1 Theoretical calculation

According to Eq. (4),  $\kappa$  can be derived experimentally from both hygroscopic growth and CCN activity measurements. Thus direct comparison of  $\kappa_{gf}$  and  $\kappa_{CCN}$  were often performed. The concentration and size dependence of  $\kappa$ , however, challenges the possibility of reaching a closure of  $\kappa$  value determined from different RH or saturation conditions.

On the basis of reliable thermodynamic models (see Appendix), we calculate the 216 theoretical  $\kappa$  values (AS, NaCl and sucrose) at RH=90% as for hygroscopic growth 217 factor measurement and at empirical critical supersaturation ( $S_{cri}$ ) as for CCN activity 218 measurement. Indeed, the retrieved  $\kappa$  is a combined presence of both concentration 219 and size dependence. As demonstrated in Fig. 3 (left panel),  $\kappa_{gf}$  decrease with

increasing particle dry diameter while  $\kappa_{\rm CCN}$  shows a contrary trend. The gap between

HTDMA and CCNC results narrows down with the increasing particle dry diameter

which is also reflected in Fig. 1. Finally, both of the predicted  $\kappa$  approach a value of

~0.48 and ~1.4 for AS and NaCl particles, respectively, while no convergence is
observed for sucrose at 200 nm.

We also notice weaker size dependence toward the larger size range. To identify 226 the size range with negligible size effects, we calculate the size dependence of  $\kappa$ 227 values as shown in the right panel of Fig. 3. The size dependence is more prominent 228 for smaller aerosols. For example, a relative variation  $(dlog \kappa/dlog D_d)$  of ~0.24 is 229 found for 10 nm AS aerosols while a  $dlog \kappa/dlog D_d$  for 50 nm aerosols is only 0.04 230 (CCN activity methods). This size effect should be specifically taking into account 231 when study the properties of nanoparticles during new particle formation and initial 232 growth.

#### 233 3.2.2 Revisit of literature data

Figure 4 summarized the comparisons of  $\kappa$  values from HTDMA and CCNC 235 measurements of field and laboratory aerosols (~50-300 nm) in literatures. Different 236 from our theoretical calculations ( $\kappa_{\rm CCN}

activity parameterizations (AP) models for AS particles, such as Aerosol Inorganics 247 Model (AIM)-based (referred as AP3) and Electrodynamic Balance method 248 249 (EDB)-based (referred as AP1, roughly corresponding to the models used in this study, see details in the Appendix)  $a_w$  parameterizations (Rose et al., 2008). For the range of 250 solute molality up to 0.3 mol kg<sup>-1</sup>, the  $a_w$  differences are on the order of ~10<sup>-3</sup>, 251 resulting in the relative differences of retrieved Se vary between 8% and 18% in the 252 253 size range of 20-200 nm (Figs. 5c and 5e), the size range where the CCNC calibration 254 commonly performed (Rose et al., 2008). Accordingly, the estimated hygroscopicity  $\kappa$ 255 is higher when applying AP3 model in CCN activity measurements compared to AP1 256 model ( $\kappa_{AS} \approx 0.66$  with AP3 and  $\kappa_{AS} \approx 0.48$  with AP1 at 200 nm). As demonstrated in 257 Figs. 3a and 6a, the deviations appear to be size dependence, varying from  $\sim 10\%$  for 258 30 nm to ~40% for 200 nm. Conversely, the influences of selecting different AP 259 models will largely cancel out when performing RH calibration in HTDMA studies 260 (Fig. 5d-e). In other words, the predicted  $\kappa$  values are not sensitive to the selected 261 thermodynamic parameterizations and Köhler models in hygroscopic growth 262 measurements.

263 Although previous studies attributed the inconsistencies between  $\kappa_{ef}$  and  $\kappa_{CCN}$  to 264 the non-ideality effects in the solution droplet and surface tension variation, etc. (Wex et al., 2009; Wu et al., 2013; Hong et al., 2014; Pajunoja et al., 2015; Zhao et al., 265 2016), we find that the key factor to explain the differences is indeed the distinct