# Peer review of "Dependence of the hygroscopicity parameter $\kappa$ on particle size,"

_Atmospheric Chemistry and Physics, 2017_

## Referee Comment (RC1) · Anonymous Referee #1 · 12 May 2017

Review on "Dependence of the hygroscopicity parameter k on particle size ..." by Wang et al., submitted to ACPD

(In here, "k" will be usedinstead of the greek kappa symbol, and "_xyz" indicates an index xyz. Abbreviations, when used, are the same as used by the authors (without introducing them again). No additional literature was added, so all citations in here can be found based on the literature list in the manuscript.)

In the study, hygroscopic particle growth and activation to cloud droplets is calculated based on models (much of that based on electrodynamic balance data (obtained for

water vapor sub saturation) and also some results from the AIM model). This is done for three different substances, AS, NaCl and sucrose. The results are used to derive k, and the retrieved values are then interpreted in the text, showing dependencies of k on solution concentration, dry particle size, the relative humidity (RH) at which k was derived (from hygroscopic growth or from droplet activation) and on the model used for calculation.

Overall, the text does contain some interesting thoughts, however, intermingled with well known ones. I got the impression that the ideas the authors want to convey were not presented clearly enough and so were lost in between well known facts. The fact that k changes with solution concentration is not new (and this change has to be there for non-ideal solutions), neither is the difference in k when derived from either hygroscopic growth or from activation (which has been acknowledged by the authors in citing numerous papers – and which they call "innate" for sucrose). A further well known point is, as already clearly said in Petters & Kreisenweis (2007), that k and surface tension (sigma_s/a) have to be applied self-consistently, meaning that when the surface tension of water was used to derive k, further use of this k in modeling has to be done using the surface tension of water, too.

It is difficult to distinguish between these well known facts and the new ideas presented in here. Also, for one topic presented as a major result, there is no real proof. It is said that deviations for k derived from measurements at sub- and supersaturated conditions (k_gf and k_CCN, respectively) will become smaller if CCN counters would be calibrated based on EDB data (referred to as AP1) instead of based on AIM (called AP3 in here), which is correct. But based only on this, is is then argued that the use of AP3 instead of AP1 for instrumental calibration explains deviations between k_gf and k_CCN as observed in atmospheric and in chamber measurements described in literature. But it is not discussed which model was used as a base for calibration for these literature data, and I guess it might not be even given in all the cited studies, and I doubt that it was always AIM. And even so, it does not become clear why the

model AP1 should be the more correct model. There might be a point here that is worth discussing, but it should be presented differently, elaborating on it more, and discussing why one model should be preferred over the other (including why smaller differences between k_gf and k_CCN should be expected).

Also, throughout the text, strong statements are made, exaggerating the results of this study and not putting them in perspective to what was known before. The text needs to be tuned down some. This is elaborated on in detail in the specific comments (together also with some positive remarks.)

Occasionally (indicated in detail below), it was difficult to understand the meaning of the text. Also, the English has to be polished throughout the text, and I indicate a number of corrections down at "technical comments", but these are, by far, not all. But solving these grammar issues alone will not affect the aforementioned issues with understanding.

Overall, the study needs major revisions before it might be considered for publication in ACP. New results should more clearly be differentiated from already known facts, more precise explanations have to be given, results from different parts of the study should be compared to enable the overall conclusions on the importance of the results, and the wording should be appropriate to what was really achieved.

Specific comments:

When I started reading the text, I was expecting to also see results from measurements you did, however, measurements were not done at all for the here presented study. It should be explicitly stated in both the abstract and the conclusions, that this study is a modelling study.

The effect of particle size is interesting. However, it is overemphasized a bit, as particles in the nano-size range < 10nm will not act as CCN in the atmosphere and hence I cannot see where the fact that there is a larger deviation in k_gf and k_CCN for these

small particles would be important. The authors should discuss this in the text and adjust all related statements.

Depending on the model used for calibration, the deviation between k_gf and k_CCN can be larger or smaller, as you argue in Fig. 6 and related text. And you suggest that AP1 models k_CCN of AS better than AP3. But you base your assumption only on the following: when k_CCN for AS is derived either from AP1 or AP3, the deviation between these two k_CCN values is roughly similar to the deviation typically observed in literature between k_gf and k_CCN for atmospheric and laboratory measurements. (You do not even mention which model was used in the calibration for these literature data). The similarity in these deviations is is not enough to decide which model is better. Parts of this comment will be repeated below, but I do say it here already: you have to describe much better how you derive the uncertainty range for deriving k_CCN, and you cannot simply "transfer" the deviation as you did it (in caption of Fig. 6). You have to include the uncertainty in the supersaturation that originates from calibrating a CCNC either with AP1 or AP3) and then from that derive the uncertainty in k_CCN for the atmospheric samples. Also, did all of the studies you cite in lines 264-266 use AP3 for CCNC calibration? Only if this were the case could your statement following these citations be made so strongly. Also, as you describe this innate deviation between k_gf and k_CCN for sucrose, discuss why you think that such a deviation should not be there for atmospheric aerosol.

It is good that it is clearly recommended that publications should always state which Köhler model they used when calibrating instrumentation to measure hygroscopic growth and droplet activation. But again, this has not been the case in the past, which makes me wonder how large the influence of the choice of model has been on derived k in past studies.

So overall: If there is this innate difference, then is the choice of model really the "key factor" to explain differences between k_gf and k_CCN (as said in the text, line 266 ff)?

You refer to "self-consistency", and I suggest you explain exactly what you mean by that in the text. I even expect that this is not the correct wording for what you want to say. Basically, what you suggest is that AP1 is better than AP3. But technically speaking, as k_gf is not influenced by the choice of model, AP3 could be used to calibrate a HTDMA and AP1 to calibrate a CCNC and k_gf and k_CCN would deviate less compared to the case when AP3 was used for the calibration of both (where the latter would be the "self-consistent" case). How does this fit to your statements about self-consistency? And anyway: Does it really happen that within one study different models are used for calibration?

I expect that with "self-consistency" you might refer to cases where k derived from measurements was then used to then derive mass fractions of substances that could possibly contribute to particle composition. I got this impression "between the lines" (e.g., lines 19-22, 124-126), but it is nowhere said very clearly. Some more examples on what you mean, explicitly, would be good. Also, if the derivation of mass fractions of different substance based on k is what you aim at, then that is one step further from only deriving k, which needs to be clearly stated.

Also make sure you always mean the same thing when referring to self-consistency: Does it really always refer to the choice of model? Or do you also refer to what already Petters & Kreidenweis (2007) said concerning the choice of surface tension (see my remark above)?

One thing I missed completely is a discussion on how precisely k has to be known for atmospheric modelling. Which uncertainty in cloud droplet concentrations is caused by the difference in k_CCN when derived based on calibration either with AP1 or AP3? Or is this resulting uncertainty so small that the use of either model is fine for atmospheric modelling?

Lines 31-34: This sentence is not clear to me. Deviations of k_CCN from what? And you give a larger deviation (in %) for the larger particle size, which seems to contradict

what you said in the sentence before. And I don't know how the deviations can indicate what you state in the second part of the sentence. This sentence needs rewriting.

Line 172: What do you mean by "their"? Also, using a different surface tension simply yields a different k, so "compensated through a change in k" is a somewhat unfit formulation in this context. Reformulate this sentence.

Line 260 ff (including the next paragraph): You state that "predicted k values are not sensitive to the selected thermodynamic parameterizations and Köhler models in hygroscopic growth measurements", but can you say that in general, or only for the two different models you compared here?

In line 272 you describe that k_gf and k_CCN are innate distinct (based on your model results for sucrose), and I wonder how this is different from the statements others you cite have made before (i.e., that there is indeed a change in non-ideality of the solution or in the surface tension etc.). I feel it should be stressed in your text that this statement of yours confirms earlier studies, moreover so as others' results are based on measurements while in your study you can, by the make of it, only obtain what was included in the model (albeit based on measurements).

Line 295: Which "near linear kappa-composition relationship" are you referring to? And what do you mean with "for estimating kappa or mass fraction of organic or inorganic compounds"? This comes from nowhere and is totally unclear. Rewrite.

Line 299: AS is AS, so strictly speaking, determining kappa for an aerosol consisting of AS particles does not make sense. - What do you mean here?

Line 303: Please indicate how much of this range is due to the choice of the model (meaning how large is that range for the same size and concentration / RH)?

Line 305-307: I get the feeling that you are referring to something done in the publications you cited here that needs to be known to understand what you mean here. It seems to indicate that in these (or other?) studies one value of k was used for calibration and another one further on (what for?). Has that really happened? If you can clearly point on that in former studies, it would help understand what you are aiming at. As it is now, there is only a vague impression created of what you could mean.

Line 309 – end of paragraph: Is this bias of 10% you are referring to originating in experimental uncertainty? And which "identical thermodynamic data" are you referring to here? For identical conditions there should not be a bias, not even 10%? And what do you mean by "inconsistent parameters", exactly? And again, how do you know which parameters are the correct ones?

Line 342-343: I guess I'm repeating myself, but it would be interesting to see how much of the inconsistency in k originates from the different contributing factors that you list in the previous sentence, to justify what you are then claiming in the next sentence.

Figure 4: Was k always determined for the same size in HTDMA and CCN measurements, and if not, which size is indicated by the symbol size? Explain in the caption or in the related text. If sizes were different, how much of the deviation between k_gf and k_CCN originated from that?

Figure 6 (b) and related text: It is not totally clear to me how the area indicating the uncertainty is derived. It can be guessed from the text in the caption, but this has to be described in much more detail in the text. And, I repeat myself, clearly state on which model the calibrations for the measurements were based in which of the studies, or for which of these studies this is not known, so that one can judge for which data the lowering in the deviation between k_gf and k_CCN can be expected.

Technical comments:

Line 47: Better use either "Kelvin and Raoult effects" or "droplet curvature and solute effects", do not use the name for one and the nature of the effect for the other.

Line 50: Consider replacing "many" with "some", also in light of what you say in the next sentence.

[Figure]

Line 68: Add and "a" between "Since" and "representative".

Line 102: Remove "ed" from "related".

Line 106: Exchange "of" with "for".

Line 149: It should be "k has" (not have). And replace "certain" with "each".

Line 175: Add a citation for the Tolman effect and shortly mention what this is.

Line 218: Exchange "is a combined presence of both" with "depends on".

Line 219: Exchange "size dependence" by "dry particle size".

Line 224: It is rather "up to" instead of "at".

Line 235: Delete the "s" in "literatures".

Lines 239-241: Difficult to understand without knowing what you describe next. Express more clearly what you mean.

Line 247: Better "for the" instead of "such as" (the latter implies that more is shown, but in Fig. 5, which you refer to here, you show only these two).

Line 248-250: Check this sentence, something went wrong here.

Line 254: Add "is" as first word in this line.

Line 255: Add "the" before "AP3" and also before "AP1".

Line 271: "In addition" is the wrong choice of word here. I recommend to start a new paragraph and let that start with "However", as the text prior to that argues that k_gf and k_CCN could wrongly be different, while the text following that explains, that these differences are innate for some substances.

Line 276: Exchange "demonstrate" with "indicate". The former is too strong of a statement here. Your results rather indicate that this could be a reason.

Line 282: You refer to Figs. 3a and 3e here. But they show a dependence on dry diameter while in the text you mention a concentration dependence.

Line 320: You use the word "experimentally"! Did you measure this? Or does this come from your calculations? Or do you cite a paper, here? Adjust the text accordingly.

Line 439: Delete the "(b)" following "Right panel:".

Line 448: It should be "models" (plural) as you talk about different ones.

Line 531-533: Check the capitalization of words – I didn't check the literature list, this one I just only ran into, so make sure this is all correct in the final version.

Fig. 4 and 6: Use the same parameters on the axis as in the text (k_gf and k_CCN), at least in Fig. 4 and Fig. 6 (b).

Fig. 6 (a): Add to the caption that the calculation was done for AS.

Fig. 6 (b): Describe somewhere (better in the text than in the caption) how exactly the upper and lower values for k_CCN were derived.

---

## Referee Comment (RC2) · Anonymous Referee #2 · 24 May 2017

The study describes discrepancies of kappa values when derived over a range of water saturation (by HTDMA or CCNC measurements) as well as for different particles sizes and solute concentrations. The authors perform calculations for three substances, ammonium sulfate (AS), sodium chloride (NaCl) and sucrose to demonstrate that varying results for kappa can be obtained based on the choice of input variables and then discuss their results together with findings from the literature. Main points of this work include that kappa needs to be carefully assessed especially for sub 10 nm particles because of kappa's size dependence; and that discrepancies in kappa can be due to calibrations that rely on different thermodynamic models.

[Figure]

While this technical study elaborates on important points that should be taken into account, e.g. specifying the thermodynamic model used for instrument calibrations, it is also a study conducted in isolation lacking discussion on its relevance. Kappa is a useful parameter to represent particle hygroscopicity which is relevant for the number concentration of cloud condensation nuclei (CCN). The CCN number concentration is size dependent and smaller particles, especially sub 10 nm, are significantly less relevant than larger particles. Based on this well-known fact the question comes up of how relevant the accuracy and precision of kappa for sub 10 nm particles is. This discussion is completely missing from the manuscript. Similarly, the title promises that this work discusses implications of varying kappa values for modeling studies. However, there is no discussion on the consequences and potential relevance of uncertainties in kappa values for any endpoint (CCN number concentration, cloud droplet number concentrations, cloud albedo, precipitation formation etc.) that might be considered in a model study.

Generally, among the points discussed the authors do not always distinguish clearly between the new contribution from their study and known aspects from previous publications. For example, a message of the paper is the concentration dependence of kappa (section 3.1) and the importance of the water activity. Already Petters and Kreidenweis (2007) have discussed this in their paper. It needs to be stated more clearly which is the new information that this work contributes and which aspects have been known beforehand. Against this background it is not fully clear which is the specific contribution the author would like to make with this work and why it is relevant.

In addition, at several occasions the authors seem to imply information without explicitly mentioning it, see specific comments section. This makes the manuscript difficult to understand and the messages are not always clear. Furthermore, the text needs to be revised for English grammar which also makes the text difficult to read. There are far too many mistakes to be pointed out at the review stage.

I recommend that this manuscript undergoes major revisions based on the above and

the more specific comments following below. It might also be worthwhile to rethink whether a "normal" scientific paper is the most appropriate type of manuscript, perhaps a "technical note" would be more appropriate.

Specific comments:

l. 21 – 24: How relevant are kappa closure studies for particles < 10 nm? Why do particles < 10 nm deserve special attention in your manuscript?

l. 99 f: The authors imply that k_gf is smaller than k_ccn in normal cases which can be inferred from the statement "Conversely, higher k_gf compared to k_ccn...". However this is not stated explicitly. The text should clearly state what the more regular observation is and then explain differences from the norm. Also, it is not clear how often k_gf > k_ccn has been found, because only one reference is provided. If there is only one study reporting this, the question is whether this is relevant information.

l. 105: Where does the information come from that kappa-composition relationships are only valid for particles > 50 nm? A reference an explanation is missing.

l. 110: What exactly in multiphase chemistry needs to be understood with regards to kappa? It is not clear what the authors imply.

l. 111: How sensitive are climate modeling studies really to the accuracy of kappa? Particle hygroscopicity is only one factor among many that determine cloud properties. This argument needs to be more elaborated to make it convincing.

l. 154: This is confusing. In line 141f it is stated that literature values were used for sigma_sol, while here it is stated that the surface tension of water is used.

l. 175: An explanation for the Tolman effect is missing as well as a reference.

l. 188f: An explanation why kappa first decreases and then reaches a plateau is missing.

l. 198ff: More elaboration of why kappa changes with size is needed given that gf

remains constant. The authors should also highlight in how far this finding is new, no references are given in this paragraph, and how it relates to the observations in the field that kappa changes with size (e.g. Good et al. 2010, Jaatinen et al. 2014). The difference with the field observations is that the chemical composition changes with size and hence kappa. In the calculations performed in this work the composition and solution concentration remains the same, still kappa changes. Jaatinen, A., Romakkaniemi, S., Anttila, T., Hyvarinen, A. P., Hao, L. Q., Kortelainen, A., ... & Laaksonen, A. (2014). The third Pallas Cloud Experiment: Consistency between the aerosol hygroscopic growth and CCN activity. Boreal environment research, 19, SS368-SS368.

l. 212: Provide references.

l. 224: An explanation is missing why there is no convergence found for sucrose at 200 nm.

l. 231: Why is it especially important to consider a changing kappa for such small particles? As long as they are not relevant for cloud formation the accuracy of the kappa value is not particularly important.

l. 237ff: The authors state that k_ccn > k_gf has mostly been attributed to potential measurement uncertainty and non-ideality of the solution but cite only one study per aspect. This is no basis for generalizing. Either more studies need to be cited or the potentially various explanations need to be named, because the authors claim to provide an alternative explanation without showing that it really is an alternative. How can the authors know that the parameterization method in all the studies was the problem? Is there enough information in each study to come to that conclusion? If yes, a table that lists the parameterization used would be helpful to get an overview. If not, more information is needed how the authors trust in their judgement that the parameterizations for the calibrations are the crucial factor for discrepancies?

l. 270f: How do the authors decide that k_gf is the norm and k_ccn the deviation? And how do the authors know that it is correct to assume that all studies used the AP3

model? As suggested above, a table of the reviewed literature on such background information would make the assumptions more transparent.

l. 273: In cases where the hygroscopic behavior is inherently different did the authors of the studies claim that closure was needed? In other words, between the lines the authors imply some misconception on hygroscopic behavior in some studies. More background information is needed for the reader to understand that.

l. 295f: What do the authors mean with the last two sentences on this page? Which are "such applications" and how is "self-consistent" defined?

l. 308: How many studies use an approach via kappa to estimate the fractions of organic and inorganic aerosol components? Is this an important application?

l. 311f: An explanation and a reference for the "nano-CCNC" are missing. Furthermore, it is not clear which particles with an unknown organic mass fraction the authors refer to. Ambient aerosol measured in the cited study, the 2.5 nm AS particles? Also, how do the authors derive the value of 50 % uncertainty in organic mass fraction in case of inconsistent parameter use?

l. 329: Again, such small particles are hardly relevant as CCN. Reframe the statement.

Figure 1: Information is missing on where the solid lines are derived from.

Figure 4: It is difficult to distinguish the symbols and their sizes in the cloud of markers. Field and lab studies could be shown in separate panels to make individual studies more visible. Also, there is only very limited information provided on the nature of the aerosol studied in the field. More information should be provided to make the comparison meaningful.

Technical comments:

l. 39: replace "for" by "as"

l. 40: studies instead of study

l. 61: delete "been"

l. 63: parameterizations

l. 68: insert "a" before representative

l. 94: or the presence "of" slightly

and many more, please take care of the grammar before any resubmission.